# Novel Liquor-Based Hot Sauce: Physicochemical Attributes, Volatile Compounds, Sensory Evaluation, Consumer Perception, Emotions, and Purchase Intent

**DOI:** 10.3390/foods12020369

**Published:** 2023-01-12

**Authors:** Ricardo S. Aleman, Jhunior A. Marcía, Ismael Montero-Fernández, Joan King, Shirin Kazemzadeh Pournaki, Roberta Targino Hoskin, Marvin Moncada

**Affiliations:** 1School of Nutrition and Food Sciences, Agricultural Center, Louisiana State University, Baton Rouge, LA 70803, USA; 2Faculty of Technological Sciences, Universidad Universidad Nacional de Agricultura, Catacamas 16201, Honduras; 3Departamento de Producción Animal y Ciencia de los Alimentos, Escuela de Ingenierías Agrarias, Universidad de Extremadura, 06007 Badajoz, Spain; 4Department of Dairy and Food Science, South Dakota State University, Brookings, SD 57007, USA; 5Department of Food Bioprocessing and Nutrition Sciences, Plants for Human Health Institute, North Carolina State University, Kannapolis, NC 28081, USA

**Keywords:** pepper, novel liquor, chili sauce, emotions, purchase intent

## Abstract

Hot sauces are popular peppery condiments used to add flavor and sensory excitement to gastronomical preparations. While hot sauce occupies a retail category well over a century old, a novel production method using liquor as the base preservative rather than traditional vinegar is now commercially available, and its uniqueness begs study. Hot sauces produced with tequila, rum, vodka, and bourbon were compared to traditional vinegar-based hot sauces concerning physicochemical properties, volatile compounds, microbiological quality, sensory scores, emotions, and purchase intent (PI). Under accelerated conditions, pH, titratable acidity (TA), water activity (Aw), viscosity, and color were analyzed weekly for 20 weeks, whereas rheological properties, coliforms and yeasts and molds were examined on weeks 1 and 20. Hexyl n-valerate, butanoic acid, 3-methyl-, hexyl ester, and 4-methylpentyl 3-methylbutanoate were found in high concentrations in the pepper mix as well as the hot sauce produced with vinegar. When compared to vinegar-based hot sauces, liquor-based hot sauces had similar Aw (*p* > 0.05), higher pH, viscosity, and L* values and lower TA, a*, and b* values (*p* < 0.05). Samples formulated with liquors increased the relaxation exponent derived from G’ values having a greater paste formation when compared to vinegar-based hot sauces. The sensory evaluation was carried out in Honduras. The liquor-based hot sauces had a significant (*p* < 0.05) impact on emotion and wellness terms. Bourbon and tequila samples had higher ratings than control samples in several wellness and emotion responses (active, energetic, enthusiastic, good, curious, pleased, stimulated, and wild). Adventurous, joyful, free, worried, refreshed, and healthy scores were not significantly (*p* > 0.05) different among treatments.

## 1. Introduction

Hot sauce, commonly called chili sauce, is a condiment used as a flavor enhancer to improve food taste due to the induction of pleasurable spicy sensations. It is frequently used in multiple gastronomic preparations in the food service industry. It is especially popular in Asian countries, as well as in the USA, Mexico, and several other Western countries [1]. In North and South America, hot sauces are mostly made of cayenne, chipotle, habanero, and jalapeño peppers. Gochujang and red peppers are popular among Korean seasonings, and Sriracha sauce is a conventional hot sauce from Thailand [2,3]. On the other hand, hot sauces are less popular in most European countries [3]. Since spices are added to different foods according to cultural aspects and individual preferences, hot sauces are often used for food items such as meats, vegetables, whole-grain, and egg dishes. Consumption habits, product information, and psychographic traits are the main variables when determining cultural differences in hot sauce type acceptance [4]. The U.S. Department of Agriculture (USDA) [5] classifies hot sauce as a hot or spicy sauce (Type I) and determines that it should be red to reddish-brown colored with a pungent odor. This type of hot sauce should also meet analytical requirements regarding the non-volatile solid’s concentration (7.5–18.0%), salt content (4.9–12.0%), acidity (2.4–5.0%), and capsaicin concentration (≥43 ppm) levels [6].

Vinegar is the predominant base ingredient used as a preservative and flavor agent in shelf-stable applications. Other ingredients such as salt, fruit, vegetables, and oil lend different flavors and characteristics, but vinegar is the common traditional base. The compounds responsible for more than 90% of the spiciness in hot sauces are capsaicin N-[(4-hydroxy-3-methoxyphenyl)-methyl]-8-methyl-6-nonenamide and dihydrocapsaicin N- [(4-hydroxy-3-methoxyphenyl) methyl]-8-methylnonanamide, both pungent alkaloids found in red peppers [7]. The industry often uses pure capsaicin extract as a low-cost method to intensify spice levels. Manufacturing operations often involve aging and fermentation processes to guarantee the desired sensory quality in the final product [8]. The color of traditional foods plays a fundamental role in the diet; it reflects culture and lifestyle, and history. It is also decisive when selecting the result for both consumers and marketing managers [9]. There is an ever-evolving interest in the food service and manufacturing industries for product diversification, the optimization of food formulations, and the development of new products with enhanced sensory properties. Hot sauces are popular food products. However, newly developed hot sauces with optimized volatile profiles and overall characteristics beyond vinegar are still lacking in the market. Similar to salt, liquor is a great flavor enhancer. Bourbon and whiskey are great flavor enhancers that are used to bring the sweetness of fruit, the richness of chocolate, and the smoky flavor to smoked or grilled meats. Similarly, rum and vodka are used to enrich the sweetness of fruits in baked goods and sauces. To date, there is no formal research regarding the effects of alcohol on physicochemical characteristics, volatile compounds, sensory attributes, consumer perception, emotions, and purchase intent of hot sauce. Therefore, in this study, we investigated liquor-based hot sauces produced with the addition of different liquors (tequila, rum, vodka, and bourbon) and compared them to a traditional product produced with vinegar. Our objective was to investigate how the unique attributes of these alcoholic ingredients would affect key attributes of the final product, including their volatile profile and, consequently, their sensory attributes.

## 2. Materials and Methods

### 2.1. Experimental Design

The pepper mix and liquor-based hot sauces produced with tequila, rum, vodka, bourbon, and vinegar (control) were produced and supplied by Swamp Dragon (Swamp Dragon., Baton Rouge, LA, USA). Figure 1 shows the process flow diagram. The hot sauce ingredients included aged peppers (Magic Plant Farms, Johnson City, TN, USA), liquor (Ultrapure, Dallas, TX, USA), salt (Morton, Chicago, IL, USA), and xanthan gum (Kelko, Sandusky, OH, USA). The specific pepper blend is proprietary information and is not shown. Each liquor-based hot sauce formulation was produced with similar proportions of the respective liquor or vinegar (a 1:1 ratio to aged pepper puree in each case) by Swamp Dragon Company. All of the liquors were diluted to 80 proof before mixing with the pepper pure. The microbial analysis, pH, water activity, viscosity, TA, and color were determined weekly from weeks 1 to 20 on accelerated conditions, whereas rheological properties were performed on weeks 1 and 20. Volatile compound and sensory analysis were performed after day 1 of storage. Three replications were performed for physicochemical analysis, microbial analysis, and volatile analysis. All of the experiments were performed in triplicate.

### 2.2. Accelerated Shelf-Life Study

For the accelerated shelf-life study, the samples were stored and kept in an incubator (Sanyo Gallenkamp Prime Incubator, Arnold Circle Cambridge, MA, USA) at 40 °C with 75% relative humidity and permanent light (15000 LX). “The rule of ten,” or Q10, a tool commonly used for the analysis of accelerated studies, is the factor by which the rate of quality decreases when the temperature is raised by 10 °C. The theoretical value Q10 = 2 was fixed [10] so that every increase of 10 °C implies a 2-fold increase in the reaction rate of the quality parameters. Thus, considering the ambient temperature around 20 °C represents Q10 of 2 and that in this study, the storage temperature of the hot sauce was 40 °C, then there are two cycles (4 days), and therefore each day of storage at 40 °C was considered as, at least, 4 days at ambient temperature.

### 2.3. Volatile Compound Profile

The volatile compounds of the hot sauce samples were extracted with headspace solid phase microextraction (HS-SPME) (2 cm–50/30 m Divinylbenzene/Carboxen/Polydimethylsiloxane). For this, 3 g of each sample was transferred to a 10 mL vial. The sorption time was set up to 30 min and the desorption time was 5 min, and the extraction temperature was 70 °C. The volatile compounds profile of samples was determined by gas chromatography-mass spectrometry (GC-MS, HP6890/5975C, Agilent Ltd., Santa Clara, CA, USA) in which nitrogen was used as the carrier gas and hydrogen gas was used for the flame ionization detector (FID) using a column Agilent DB-5, 60 m and 0.32 mm diameter. The GC-MS was programmed using the conditions described [11]. The procedure was followed as 60 °C (initial), 300 °C (final), gradient: 10 °C/min for 5 min. The injection was conducted in the splitless mode (1 min) at 280 °C. The MS ionization potential was 70 eV, the ionization current was 350 μA, and the ion source and transfer line temperature were set at 230 and 280 °C, respectively. The mass spectrometry was set up in Full Scan (50 to 550 *m*/*z*). The identification of volatile compounds was performed by comparing the peaks with the values provided by the NIST mass spectrum library with the help of the MASSLAB program.

### 2.4. Physicochemical Parameters during Storage

The pH was measured by using a Thermo Orion 3 Star pH Benchtop Meter (Fisher Scientific, Pittsburgh, PA, USA). Titratable acidity (TA) was performed according to [12] based on Equation (1). Briefly, 9 g of the sample was mixed with 9 mL of distilled water and 0.5 mL of phenolphthalein, and the samples were titrated with 0.1 N NaOH (Equation (1)). The equivalence endpoint was also measured (8.3 pH). The water activity (Aw) of the hot sauce was estimated with a water activity meter (Hygrolab, Rotronic, Hauppauge, NY, USA). The apparent viscosity of hot sauce was analyzed by using a Brookfield rotational DV-II viscometer (Brookfield Engineering Lab Inc., Stoughton, MA, USA) equipped with an LV #1 spindle (20 rpm, helipath). The CIELAB color parameters L*, a*, and b* were determined with a colorimeter (LabScan Model-5100, Hunter Lab Inc., Reston, VA, USA).
TA = (volume of titrant × normality of titrant × 90)/(Weight of sample × 1000) × 100(1)

### 2.5. Rheological Measurements

The rheological properties of the hot sauce samples were studied using a controlled-stress rheometer (R/S, Brookfield Instruments Co., Inc., Stoughton, MA, USA) coupled with a cone spindle (V-20/40). The samples were measured using 16-ounce Reynolds RDC212-Del-Pak Combo-Pak containers (Alcoa, Inc., Pittsburgh, PA, USA). Three types of analyses were performed, including frequency ramp (0.01–10 Hz), steady shear test (0.01–100 1/s), and dynamic stress sweep (0.1–100 Pa). All of the analyses were conducted at 25 °C. The frequency sweep analyses were fixed at 1 Pa, and the stress sweep measurements were fixed at 1 Hz. The Herschel–Bulkley and power law model (Equations (2) and (3)) was used to describe the rheological behavior of the hot sauces.
t = t_0_ + ky^n^(2)
where t is the shear stress [Pa], t_0_ the yield stress [Pa], k the consistency index [Pa × sn], y the shear rate [1/s], and n the flow index [dimensionless].
σ = ky^n^(3)
where k is the flow consistency index (Pa × sn), y is the shear rate or the velocity gradient perpendicular to the plane of shear (1/s), and n is the flow behavior index (dimensionless).

### 2.6. Sensory Evaluation

This study was approved by the Honduran Association of Physicians-Nutritionists (ASOHMENU) with form # AS-ASHOMENU-007-2022. The sensory analysis was performed using a 9-point hedonic scale [13] (1 = extremely dislike and 9 = extremely like) with 200 non-trained consumers, consisting of faculty, staff, and students at UNAG (Universidad Nacional de Agriculture), Honduras. The sensory evaluation was performed and conducted in a partitioned booth. The samples were analyzed using a three-digit code after one day of storage. The sensory characteristics studied included color, aroma, flavor, spiciness, and overall linking. EsSense (Active, adventurous, energetic, enthusiastic, free, good, joyful, pleased, wild, and worried) and wellSense terms (Alert, curious, healthy, refreshed, and stimulated) terms were obtained from the review [14], and all of the terms were evaluated using a 5-point hedonic scale [13] (1 = extremely dislike and 5 = extremely like). Purchase intent was analyzed and evaluated using a yes/no scale. The panelists were asked about gender and product familiarity [14]. Each panelist received two samples consisting of a bottle of hot sauce (about 100 mL) for each sample, unsalted crackers (Nabisco, Northfield, IL, USA), and a cup of water (Nestle Waters, Greenwich, CT, USA). The sensory study plan was executed using a Counterbalance Design (t = 5, b = 2, k = 40) and analyzed one day after the production of samples. t = 5 relates to the number of treatments, including the control, b = 2 means that a panelist only tested two samples and k = 40 means that each block was tasted by 40 people, according to the methodology described by Borgogno et al., 2017 [15].

### 2.7. Microbiological Analysis

The hot sauce samples were tested for coliforms, yeast, and molds using Petrifilm (3M Company, St. Paul, MN, USA). These analyses were performed by preparing serial dilutions in peptone water (0.1% *w*/*v*) and plating the samples in duplicate for coliforms, yeasts, and molds. The samples were aerobically incubated for 24 h at 32 °C (coliforms) or 72 h at 22 °C (yeasts and molds) before enumeration.

### 2.8. Statistical Analysis

The effects of the treatment (hot sauces prepared with different liquor types or control) and storage time (weeks) on the investigated parameters (pH, titratable acidity, water activity, viscosity, and color) were analyzed using ANCOVA with PROC MIXED and a two-factor factorial analyses in a randomized block design for the physicochemical and microbiological analyses. Bonferroni (Dunn) was used to determine significant differences for main effects (liquor type and storage time), interaction effect (liquor type × storage time), and hedonic responses. One-way analysis of variance (ANOVA) and Bonferroni (Dunn) test were conducted to analyze sensory hedonic scores. The non-parametric test Q Cochran was applied to determine statistical differences in purchase intent. Logistic regression was used to determine factors impacting purchase intent. A 5% degree of difference (*p* < 0.05) was applied to all of the tests. The experiments were conducted at least in triplicate unless noted.

## 3. Results and Discussion

### 3.1. Volatile Compound Profile of Hot Sauces

The aromatic profile of hot sauces prepared with the addition of liquor showed a more diverse compound profile compared to hot sauces prepared with vinegar. Eighteen different compounds were detected for tequila, bourbon, rum, and vodka, as opposed to just 15 aromatic compounds in the control sample.

The analyses were carried out by headspace solid-phase micro-extraction with gas chromatography and mass spectrometry (HS-SPME-GC-MS), which is an alternative volatile extraction system to the traditional methods (liquid–liquid extraction, solid-phase extraction, dynamic headspace, or vacuum distillation), which are more expensive, require a longer time to complete and are more labor-demanding. Terpenes such as bicyclo [2.2.1]heptan-2-one, (1R)-4,7,7- trimeth with its stereoisomer 1S which are reported in turmeric plants [16] and 2-acetyl-4,4-dimethyl-cyclopentane-2-enone were detected in our hot sauce samples. In addition, 6-methylhept-4-en-1-yl-3-methylbutanoate, 6-methylhept-4-en-1-yl-2-methylbutanoate, cis-(-)-2,4a,5,6,9a-Hexahydro-3,5,5,9-tetramethyl-benzocycloheptene previously reported in habanero-type peppers Capsium chinense; [17,18] and 5-hydroxy-6-methoxy-8-[(4-amino-1-methylbutyl)amino] quinoline trihydrobromidewere compounds generally found in all of the samples.

To evaluate the main differences between the samples, Table 1 shows the most predominant compounds in each experimental sample with their respective retention time and percentage area. Hexyl n-valerate, butanoic acid, 3-methyl-, hexyl ester, and 4-methylpentyl 3-methylbutanoate were found in high concentrations in the pepper mix as well as the hot sauce produced with vinegar. The hexyl n-valerate was detected by authors previously [19] in tabasco peppers, while butanoic acid and 3-methyl- hexyl ester are major volatile compounds of habanero peppers [20]. The sensory attributes provided by the different volatile compounds identified were: dimethyl ether, faint ethereal odor; ethanol has an alcohol odor; butanoic acid 3-methyl-hexyl-ester present the odor of wine and ethyl-9-decenoate, 4-methylpentyl-3-methylbutanoate, and hexyl n-valerate present with a green and fruity odor.

Overall, the hot sauce prepared with vinegar had a different aromatic compound profile compared to alcoholic sauces. While dimethyl ether and ethanol were the most relevant volatile compounds found in hot sauces formulated with rum, tequila, vodka, and bourbon, these compounds were not detected in the control sample. Our results agree with previous studies [21,22], which found dimethyl ether and ethanol in tequila, rum, and bourbon. In addition, urea and 1,1-dimethyl-hydrazine, were major compounds in the hot sauce formulated with vinegar but not found in major amounts in the alcoholic hot sauces or the pepper mix (Table 1). This corroborates with a previous report [23] showing that urea results from the fermentation process of vinegar. Interestingly, tequila hot sauce’s aromatic profile showed important differences when compared to all three alcoholic sauces. The aromatic compound bicyclo [2.2.1]heptan-2-one, 4,7,7- trimethyl-, (1R)- and its stereoisomer 1S were both found in major concentrations in the pepper mix, and ethyl 9-decenoate was among the top five compounds found in the tequila sauce. However, the same trend was not observed for bourbon, vodka, or rum hot sauces. Instead, these three alcoholic sauces presented 4-methylpentyl -3-methylbutanoate, butanoic acid, 3-methyl-hexyl ester, and hexyl n-valerate among the most prevalent aromatic compounds in their profile.

### 3.2. Physochemical Attributes during Storage

The pH of the hot sauces was within 3.10–3.85 (Figure 2A). This acidic pH range is commonly found in hot sauce products and allows for a desirable microbiologically safe condition [24] (Chung, Jorgensen, and Price, 1988; USDA, 2005). Within the same formulation, the pH level did not vary during the 20 weeks of storage, but the type of liquor used to produce the hot sauce influenced the pH (*p* < 0.05). In this regard, hot sauces made with tequila had similar pH compared to the control (*p* > 0.05), whereas vodka, bourbon, and rum sauces had significantly higher pH (*p* ˂ 0.05) than the vinegar-based control sample. Tequila has a pH of around 3.2, and it is the most acidic among the liquors [25], which explains the lower pH observed for tequila-based sauce. On the other hand, mash liquors, such as bourbon and vodka, have higher pHs of around 4, while sugar liquors, such as rum, have a pH of approximately 5 [26].

Similar behavior was observed for titratable acidity results (Figure 2B). The control samples prepared with vinegar had significantly (*p* > 0.05) higher acidity than those produced with bourbon, vodka, rum, and tequila. The pH of hot sauce is an estimate of free hydrogen ions, whereas the TA is an estimation of total hydrogen ions. The increasing TA values are related to the decrease in pH values, having a negative correlation. For all samples, the acidity values were similar to previously reported commercial hot sauces ranging from 0.6% to 1.8% [10].

The water activity (Aw) consistently comes into play as it affects the microbiological stability of food matrices [27]. The water activity values were stable during the 20 weeks of storage under accelerated conditions, and the type of liquor did not impact the water activity values. All of the samples had water activity between 0.91–0.96, with a percentage error of 5% (Figure 3).

With regard to apparent viscosity, the liquor type and the storage time significantly affected viscosity (*p* ˂ 0.05, Figure 3B). The viscosity of the hot sauce samples decreased for all samples during storage. The control samples had the lowest viscosity values, whereas the hot sauces containing rum, vodka, and bourbon reported the highest values. Xanthan gum (XG) plays a critical role in viscosity. XG in strong acidified conditions reported lower viscosity when compared to weakly acidified aqueous media [28]. Possibly, XG at lower pH adopts more stable helical conformations that lower the size of hydrodynamic volumes [29]. Under strongly acidified conditions, there is a loss of pyruvate groups that are associated with the reduction in molecular weight, leading to a shorter chain [30]. In addition, ethanol inclusion in XG/water did not lead to a hydrodynamic volume of macromolecules in dilute solutions [31].

### 3.3. Instrumental Color

The liquor type and the storage time significantly affected the color parameters of hot sauces (*p* ˂ 0.05). During accelerated conditions, the L* (27.45–25.78) values slightly decreased and the a* (31.34–33.56) and b* (26.84–36.72) values increased for all hot sauce samples. The changes in L*, a*, and b* under the accelerated conditions are possibly caused by Maillard reactions, which could be caused by the carbohydrates and proteins present in peppers [32]. For the L* and a* values, the control and hot sauce with the tequila samples had the highest values, whereas the hot sauces containing rum reported the lowest values (Figure 4A,B). For the b* values, the hot sauce with tequila and vinegar had the highest values, while the hot sauces incorporating rum showed the lowest values (Figure 4C). Its phenomenon was possible because the Maillard reaction kinetics are slower at lower pH [33]. In addition, the degradation of lycopene derived from the peppers could also contribute to the color changes in hot sauce. It was found that the change of lycopene influenced the color attributes of hot pot sauce during 120 day-storage [31].

### 3.4. Rheological Properties

The stress sweep results indicated that the hot sauce with vinegar had the lowest yield stress (τc) values (Table 2) for weeks 0 and 20, and the τc value decreased over time for all samples. In other words, these results showed that the control samples needed the lowest stress to generate structure deformation and had the lowest stability. On the other hand, hot sauces made with rum had the highest τc values, and they needed higher stress to distort their initial structure. This phenomenon can be associated with the favorable deformation of xanthan gum in hot sauce at lower pH, leading to a weaker structure [34]. In addition, ethanol inclusion at 30% in XG/water media did not affect viscosity parameters [31] and possibly did not affect XG gel formation properties in the hot sauce matrix. XG increases electrostatic, H-bonding, and hydrophobic interactions [35], and the lower n’ and n’’ values in the vinegar hot sauce could be due to the destruction by acetic acid of the hydrophobic bonds, in which pyruvate groups are involved.

### 3.5. Coliform Counts and Yeast and Mold Counts

Coliforms, yeasts, and molds were not detected in the hot sauce samples during 20 weeks of storage. This indicates that the formulation of the products did not affect their microbiological stability, and they were safe to be consumed in the timeframe investigated in this study (20 weeks of storage). In other words, coliform counts and yeast and mold counts were not present during 140 days at accelerated conditions and possibly at 560 days under regular conditions (Q10 prediction).

### 3.6. Sensory Evaluation

When compared to the vinegar-based hot sauces, the liquor-based hot sauces had higher sensory scores for aroma, flavor, spiciness, and overall liking but lower scores for color (*p* < 0.05). The formulations with tequila, bourbon, and rum had significantly higher purchase intent scores compared to the control (*p* < 0.05), and bourbon was the absolute favorite in this aspect, with a PI of 87.75% among consumers included in this study (Table 3). Indeed, alcoholic flavors and aromas may well be more appealing to consumers’ acceptance than vinegar flavors and aromas.

### 3.7. Consumer’s Emotions and Wellness Perception of Hot Sauces

In this study, it was observed that the liquor-based hot sauces had a significant (*p* < 0.05) impact on emotion and wellness terms (Table 4). The bourbon and tequila samples had higher ratings than the control samples in several wellness and emotion responses (active, energetic, enthusiastic, good, curious, pleased, stimulated, and wild). Adventurous, joyful, free, worried, refreshed, and healthy scores were not significantly (*p* > 0.05) different among the treatments.

These results were probably caused by the higher spiciness, flavor, aroma, and overall liking scores in the bourbon and tequila samples (Table 3). Previous studies [12] reported similar emotions (active, alert, energetic, enthusiastic, free, focused, good, healthy, interested, joyful, pleased, refreshed, satisfied, stimulated, wild, adventurous, and curious) in hot sauces with the visual cue analysis. Red is related to emotions, including energy, heat, power, passion, strength, and stimulation [36]. Consumers with different cultures may perceive food products differently, and therefore, it may affect emotion and wellness differently [37].

### 3.8. Purchase Intent of Hot Sauces

The addition of liquor in the formulation of hot sauces was analyzed by showing the effect of sensory properties, overall liking, emotions, and well senses on PI (Table 5).

The PIs for the bourbon, tequila, and rum samples (87.75%, 73.5%, and 65%, respectively) were higher than the control and vodka samples (55.25% and 58%, respectively) (Table 2). This tendency overlapped with the spiciness, flavor, aroma, and overall liking scores (Table 2). Logistic regression was applied to determine attributes critical for predicting PI (Table 5). Overall liking, aroma, flavor, spiciness, familiarity, gender, wild, energetic, stimulated, and curious were significant predictors with odds ratio values of 1.364, 1.072, 1.001, 1.207, 1.177, 1.107, 1.227, 1.028, 1.232, and 1.256, respectively. The consumers’ willingness to purchase hot sauce was not only influenced by liking, gender, and familiarity but also affected by sensory aspects such as flavor, spiciness, and aroma. Ngoenchai et al. (2019) [14] reported that wild and familiarity affected PI in chill peppers with the visual cue analysis.

## 4. Conclusions

The liquor-based hot sauces had different pH, TA, viscosity, color, and rheological properties when compared to the vinegar-based samples, providing evidence for novel flavor and aroma experiences against a vinegar-based category of products. A more diverse volatile profile of hot sauces formulated with tequila, bourbon, rum, and vodka was detected. In addition, whereas dimethyl ether and ethanol were the most relevant volatile compounds found in liquor-based hot sauces, they were not detected in the control sample, again suggesting a product altogether different from anything else in its product category. Because of the complex aroma and desirable flavor characteristics of these alcoholic ingredients, liquor-based hot sauces received higher sensory scores. Enhanced overall liking, aroma, flavor, and spiciness, were perceived, and the bourbon and tequila-derived hot sauces had higher ratings in important wellness and emotion responses (active, energetic, enthusiastic, good, curious, pleased, alert, stimulated, and wild). Similarly, the purchase intent was higher for the liquor-based sauces compared to the control, except for the vodka hot sauce. Formulating hot sauce with liquor is a promising strategy to deliver novel peppery products with promising market acceptability for the food and its related service industries. For future research, consumer familiarity, expected heat intensity, preference mapping techniques, and conjoint analysis should be addressed to understand consumer behavior.

## Figures and Tables

**Figure 1 foods-12-00369-f001:**
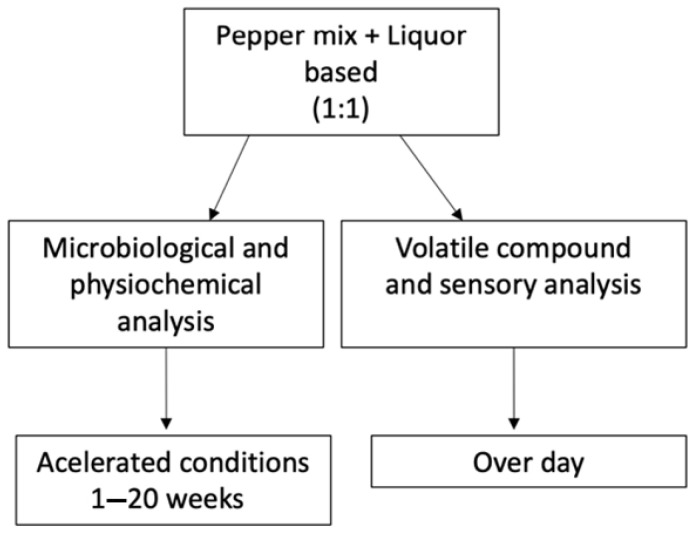
Process Flow Diagram.

**Figure 2 foods-12-00369-f002:**
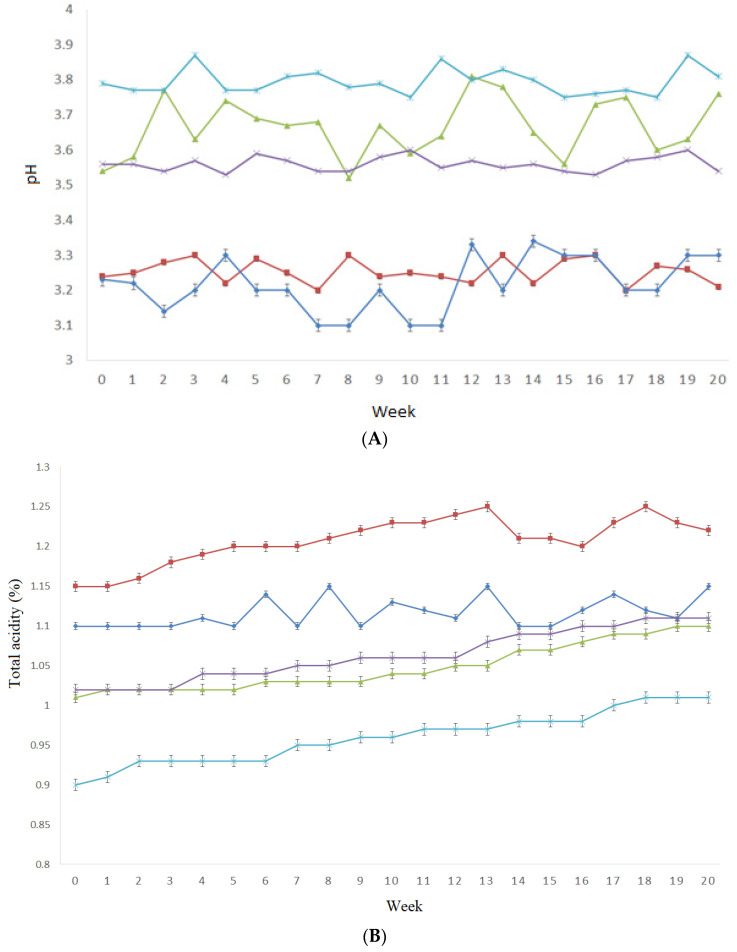
pH (**A**) and titratable acidity (**B**) of hot sauces prepared with tequila (dark blue line), bourbon (green line), vodka (purple line), rum (light blue line), and control (brown line) during 20 weeks of storage under accelerated conditions.

**Figure 3 foods-12-00369-f003:**
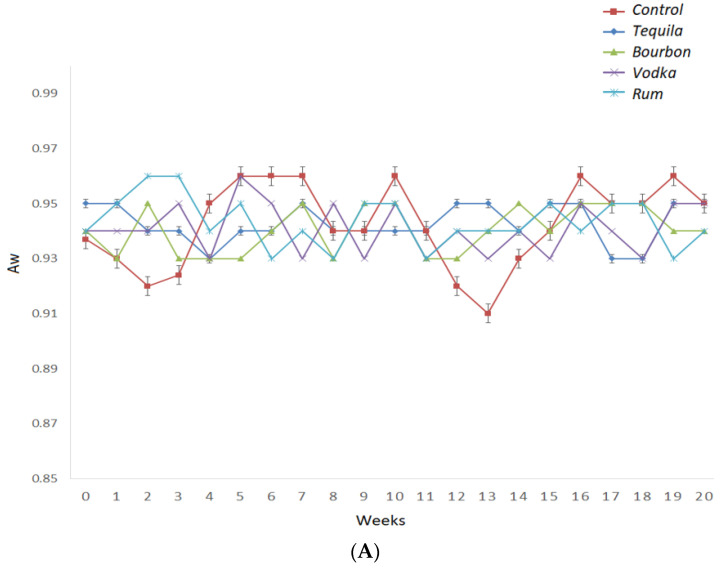
Water activity (**A**) and viscosity (**B**) of hot sauces prepared with tequila (dark blue line), bourbon (green line), vodka (purple line), rum (light blue line), and control (brown line) during 20 weeks of storage under accelerated conditions.

**Figure 4 foods-12-00369-f004:**
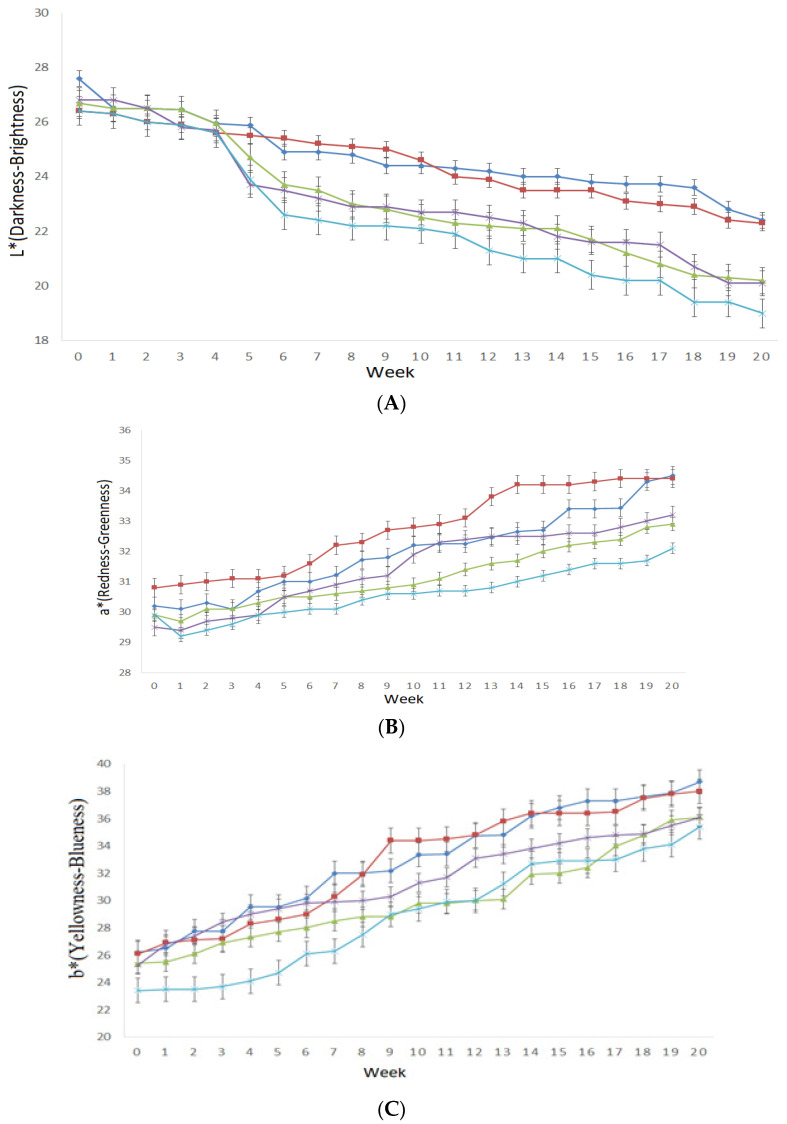
Instrumental color CIELAB parameters L* (**A**), a* (**B**), and b* (**C**) of hot sauces prepared with tequila (dark blue line), bourbon (green line), vodka (purple line), rum (light blue line), and control (brown line) during 20 weeks of storage under accelerated conditions.

**Table 1 foods-12-00369-t001:** Major volatile compounds detected in hot sauces formulated.

Vinegar (Control)	RT	Area (%)	Tequila	RT (min)	Area (%)	Bourbon	RT (min)	Area (%)	Vodka	RT (min)	Area (%)	Rum	RT (min)	Area %	Pepper Mix	RT (min)	Area %
4-Methylpentyl 3-methylbutanoate	15.5	42.1	Dimethyl ether	1.4	23	Dimethyl ether	1.4	46.7	Dimethyl ether	1.4	47.3	Dimethyl ether	1.4	52.6	Bicyclo [2.2.1]heptan-2-one, 4,7,7- trimethyl-, (1R) &(1S)	19.9	27.1
Butanoic acid, 3-methyl-, hexyl ester	15.5	42.1	Ethanol	1.4	23	Ethanol	1.4	46.7	Ethanol	1.4	47.3	Ethanol	1.4	52.6	2-Acetyl-4,4-dimethyl-cyclopent-2-enone	19.9	27.1
Hexyl n-valerate	15.5	42.1	Ethyl 9-decenoate	19.9	20	4-Methylpentyl 3-methylbutanoate	18.5	15.4	4-Methylpentyl 3-methylbutanoate	18.5	15.4	4-Methylpentyl 3-methylbutanoate	18.5	15.4	4-Methylpentyl 3-methylbutanoate	18.5	22.5
1,1-dimethyhydrazinel	1.5	28.4	Bicyclo [2.2.1]heptan-2-one, 4,7,7- trimethyl-, (1R)	19.9	20	Butanoic acid, 3-methyl-, hexyl ester	18.5	15.4	Butanoic acid, 3-methyl-, hexyl ester	18.5	15.4	Butanoic acid, 3-methyl-, hexyl ester	18.5	15.4	Butanoic acid, 3-methyl-, hexyl ester	18.5	22.5
Urea	1.5	28.4	Bicyclo [2.2.1]heptan-2-one, 4,7,7- trimethyl-, (1S)	19.9	20	Hexyl n-valerate	18.5	15.4	Hexyl n-valerate	18.5	15.4	Hexyl n-valerate	18.5	15.4	Hexyl n-valerate	18.5	22.5

Treatment: vinegar (control), tequila, bourbon, vodka, and rum, and in the pepper mix used to produce sauce samples. Aromatic compounds are presented in order of percentage area (from the highest to the lowest) in each hot sauce sample. RT = retention time (min).

**Table 2 foods-12-00369-t002:** Rheological parameters determined from amplitude sweep and steady shear flow experiments on Herschel–Bulkley and Power Law models for flow curves and dynamic viscoelasticity of hot sauces formulated with vinegar (control), tequila, bourbon, vodka, and rum.

Sample	Flow Properties				τc (Pa) *	Dynamic Viscoelasticity	
	K (Pa·sn)	n	τ_0_	R^2^		n”	n’
T1_week0_	11.18 ± 0.10 ^a^*	0.25 ± 0.01 ^a^*	0.95 ± 0.03 ^a^*	0.99	0.89 ± 0.01 ^a^*	0.35 ± 0.11 ^a^*	0.63 ± 0.02 ^a^*
T1_week20_	8.47 ± 0.07 ^a^*	0.39 ± 0.02 ^a^*	0.71 ± 0.05 ^a^*	0.99	0.56 ± 0.02 ^a^*	0.40 ± 0.07 ^a^*	0.77 ± 0.12 ^a^*
T2_week0_	10.34 ± 0.05 ^a^*	0.23 ± 0.03 ^a^*	0.87 ± 0.07 ^a^*	0.97	0.73 ± 0.03 ^b^*	0.25 ± 0.05 ^b^*	0.55 ± 0.07 ^b^*
T2_week20_	9.30 ± 0.13 ^a^*	0.37 ± 0.05 ^a^*	0.75 ± 0.04 ^a^*	0.99	0.45 ± 0.05 ^b^*	0.45 ± 0.10 ^b^*	0.61 ± 0.04 ^b^*
T3_week0_	11.23 ± 0.15 ^a^*	0.24 ± 0.01 ^a^*	0.96 ± 0.03 ^a^*	0.99	0.84 ± 0.05 ^a^*	0.39 ± 0.07 ^a^*	0.59 ± 0.06 ^a^*
T3_week20_	8.45 ± 0.10 ^a^*	0.33 ± 0.04 ^a^*	0.82 ± 0.07 ^a^*	0.95	0.33 ± 0.04 ^c^*	0.45 ± 0.13 ^a^*	0.80 ± 0.02 ^a^*
T4_week0_	9.06 ± 0.08 ^ab^*	0.26 ± 0.07 ^ab^	0.72 ± 0.06 ^ab^*	0.99	0.76 ± 0.07 ^b^*	0.30 ± 0.08 ^b^*	0.50 ± 0.05 ^b^*
T4_week20_	8.34 ± 0.13 ^a^*	0.39 ± 0.04 ^a^*	0.77 ± 0.05 ^a^*	0.99	0.34 ± 0.04 ^c^*	0.34 ± 0.04 ^b^*	0.67 ± 0.02 ^b^*
C_week0_	8.73 ± 0.17 ^b^*	0.34 ± 0.03 ^b^*	0.67 ± 0.04 ^b^*	0.97	0.54 ± 0.03 ^c^*	0.20 ± 0.07 ^c^*	0.41 ± 0.08 ^c^*
C_week20_	7.05 ± 0.13 ^b^*	0.45 ± 0.05 ^b^*	0.54 ± 0.07 ^b^*	0.99	0.25 ± 0.05 ^d^*	0.25 ± 0.05 ^c^*	0.50 ± 0.03 ^c^*

Average of three replicates. Values with different letters in columns are significantly different (*p* < 0.05) within columns. τ_0_ = Yield stress, n = flow index, *K* = consistency coefficient, n’ = relaxation exponent of G’, n” = relaxation exponent of G”. letters = differences between samples at days 1 and 42 separately. Rum = T1, Vodka = T2, Bourbon = T3, Tequila = T4, Vinegar = C. * indicates a significant (*p* < 0.05) difference between days 1 and 42.

**Table 3 foods-12-00369-t003:** Sensory scores and intention of purchasing hot sauces formulated with vinegar (control), tequila, bourbon, vodka, and rum.

Attribute	Hot Sauce Sample				
	Control	Tequila	Bourbon	Vodka	Rum
Color	7.01 ± 1.4 ^a^	6.97 ± 1.5 ^a^	6.95 ± 1.3 ^a^	6.92 ± 1.9 ^b^	6.88 ± 1.7 ^b^
Aroma	6.43 ± 1.5 ^b^	7.15 ± 2.0 ^a^	7.28 ± 1.5 ^a^	7.11 ± 2.1 ^a^	7.22 ± 1.9 ^a^
Flavor	6.05 ± 1.7 ^b^	6.50 ± 2.1 ^a^	6.75 ± 1.8 ^a^	6.39 ± 1.3 ^a^	6.47 ± 1.1 ^a^
Spiciness	6.09 ± 1.1 ^b^	6.86 ± 1.1 ^a^	6.90 ± 1.1 ^a^	6.64 ± 1.2 ^a^	6.79 ± 1.2 ^a^
Overall liking	6.15 ± 1.4 ^b^	6.72 ± 1.5 ^a^	6.95 ± 1.8 ^a^	6.47 ± 1.6 ^a^	6.55 ± 1.1 ^a^
The intention of purchase (%)	55.25 ^d^	73.50 ^b^	87.75 ^a^	58.00 ^d^	65.00 ^c^

Mean ± standard deviation of results. Frequencies based on a “yes/no” scale (Intention of purchase). Mean values in the same row followed by different letters (^a^, ^b^, ^c^, ^d^) are significantly different (*p* < 0.05) using the Tukey test.

**Table 4 foods-12-00369-t004:** Mean consumer emotions and well senses scores of hot sauces formulated with vinegar (control), tequila, bourbon, vodka, and rum.

Attribute	Hot Sauce Type				
	Control	Bourbon	Tequila	Rum	Vodka
**EsSense ProfileTM Terms**					
Active	3.27 ± 1.21 ^b^	3.78 ± 1.45 ^a^	3.55 ± 1.07 ^ab^	3.47 ± 1.09 ^b^	3.44 ± 1.10 ^b^
Energetic	3.03 ± 1.10 ^c^	3.86 ± 1.23 ^a^	3.62 ± 1.34 ^ab^	3.45 ± 1.41 ^b^	3.47 ± 1.27 ^b^
Enthusiastic	3.07 ± 1.34 ^c^	3.48 ± 1.55 ^b^	3.94 ± 1.07 ^a^	3.55 ± 1.47 ^bc^	3.47 ± 1.12 ^b^
Good	3.12 ± 1.14 ^c^	3.78 ± 1.32 ^a^	3.73 ± 1.21 ^a^	3.77 ± 1.18 ^a^	3.46 ± 1.19 ^b^
Pleased	3.03 ± 1.05 ^b^	3.66 ± 1.33 ^a^	3.75 ± 1.07 ^a^	3.70 ± 1.25 ^a^	3.85 ± 1.40 ^a^
Adventurous ^NS^	1.37 ± 1.46 ^a^	1.58 ± 1.55 ^a^	1.27 ± 1.35 ^a^	1.57 ± 1.37 ^a^	1.44 ± 1.46 ^a^
Joyful ^NS^	3.34 ± 1.11 ^a^	3.25 ± 1.23 ^a^	3.45 ± 1.36 ^a^	3.17 ± 1.17 ^a^	3.30 ± 1.24 ^a^
Free ^NS^	3.56 ± 1.03 ^a^	3.65 ± 1.25 ^a^	3.77 ± 1.29 ^a^	3.68 ± 1.05 ^a^	3.58 ± 1.11 ^a^
Wild	3.01 ± 1.16 ^c^	3.95 ± 1.29 ^a^	3.79 ± 1.05 ^a^	3.49 ± 1.11 ^b^	3.33 ± 1.17 ^bc^
Worried ^NS^	2.34 ± 1.05 ^a^	1.98 ± 1.01 ^a^	1.95 ± 0.92 ^a^	2.05 ± 0.83 ^a^	1.90 ± 0.94 ^a^
**WellSense Profile^TM^ terms**					
Stimulated	2.18 ± 1.32 ^c^	2.88 ± 1.04 ^a^	2.70 ± 1.16 ^a^	2.83 ± 1.10 ^a^	2.37 ± 1.28 ^b^
Alert	2.38 ± 1.23 ^b^	2.87 ± 1.12 ^a^	2.56 ± 1.32 ^ab^	2.44 ± 1.05 ^ab^	2.50 ± 1.07 ^ab^
Healthy ^NS^	2.37 ± 1.04 ^a^	2.42 ± 1.21 ^a^	2.40 ± 1.23 ^a^	2.32 ± 1.22 ^a^	2.35 ± 1.07 ^a^
Curious	2.01 ± 1.32 ^c^	2.99 ± 1.05 ^a^	2.87 ± 1.27 ^a^	2.32 ± 1.35 ^b^	2.35 ± 1.28 ^b^
Refreshed ^NS^	2.11 ± 1.23 ^a^	2.05 ± 1.14 ^a^	2.15 ± 1.08 ^a^	2.20 ± 1.01 ^a^	2.13 ± 1.22 ^a^

Mean ± standard deviation base on a 5-point scale. ^a^, ^b^, ^c^: Mean values in the same row followed by different letters are significantly different (*p* < 0.05). ^NS^ = Not significant regarding emotional responses among treatments.

**Table 5 foods-12-00369-t005:** The odds ratio for predicting purchase intent with sensory properties, emotions, senses, and non-sensory factors of hot sauces formulated with vinegar (control), tequila, bourbon, vodka, and rum.

Attributes	Before	
	Pr > *X*^2^	Odds Ratio
Overall liking	<0.001	1.364
Gender	0.035	1.107
Familiarity	0.027	1.177
Color	0.075	0.97
Aroma	0.038	1.072
Flavor	0.048	1.001
Spiciness	0.026	1.207
Active	0.643	0.364
Energetic	0.042	1.028
Enthusiastic	0.534	0.232
Good	0.110	0.734
Pleased	0.087	0.957
Adventurous	0.158	0.695
Joyful	0.103	0.709
Free	0.095	0.923
Wild	0.014	1.227
Worried	0.156	0.701
Stimulated	0.022	1.232
Alert	0.345	0.304
Healthy	0.157	0.634
Curious	0.007	1.256
Refreshed	0.150	0.668

The results were obtained by logistic regression analysis, using a full model of sensory properties, overall liking, emotions, and well senses. Analysis of maximum likelihood estimates was used to obtain parameter estimates. The significance of parameter estimates was based on the Wald *X*^2^ value at *p* < 0.05.

## Data Availability

Data is contained within the article.

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
