# Peer review of "Novel Liquor-Based Hot Sauce: Physicochemical Attributes, Volatile Compounds, Sensory Evaluation, Consumer Perception, Emotions, and Purchase Intent"

_foods, 2023, doi:10.3390/foods12020369_

Round 1

Reviewer 1 Report

Included in the attached file

Reviewer 2 Report

The abstract does not mention that sensory evaluation was conducted in Honduras. The literature review should discuss how consumer preferences for heat in foods vary among nations or cultures.

The formulations should be described in more detail, including the name and location of ingredient suppliers. Were all liquors adjusted to the same percentage of alcohol (proof)?

Lines 79-80 state that sensory evaluation was done after one day of blending, but lines 143-144 say that testing occurred after one week of storage. Which date is correct? How did you determine how long the sauces should be stored before evaluation? Are aged sauces preferred to freshly-made sauces?

Why was a 9-point hedonic scale used for the sauces and a 5-point scale used for emotional terms? Why were demographic traits not described in the methods? Please add a table showing the demographic characteristics of test participants. It is important to know how often sensory test participants used hot sauce or other spicy condiments before the test.

Please spell out the abbreviations in lines 151-153. What is meant by f day?

Was there a measurement error in the water activity data? There was considerable variation.

Why are some terms italicized in Table 5?

Reviewer 3 Report

Novel Liquor-Based Hot Sauce: Physicochemical Attributes, Volatile Compounds, Sensory Evaluation, Consumer Perception, Emotions, and Purchase Intent.

Some problem shoud be solved.

Review Comments:

1、  The abstract does not state what value the study of the unique properties of Liquor-based chutneys provides, only the background, methodology, content, and results of the study, which should be supplemented with its research implications.

2、  The background information on the hot sauce is adequate and raises the status of issues such as the lack of novel, non-volatile characteristics of hot sauce products, but does not clearly answer this and should be supplemented with targeted opinions.

3、  The paper characterizes vinegar-based hot sauces and Liquor-based hot sauces by determining the number of indicators such as physicochemical properties, volatile compounds, microbiological quality, hedonic scores, emotions, and purchase intent to illustrate the unique properties of wine-based hot sauces. The research methodology is presented and the article is well laid out and structured to illustrate the unique properties of wine-based hot sauces. However, the results of the paper are imperfect, and what advantages its unique attributes have, for example, how volatiles affect key product attributes are not specifically written, and it is recommended that the flavor characteristics of volatiles be added.

4、  The chart is clear and reliable, but the literature citation is old, less in the past 5 years, and the sample reagents, instruments, and equipment used in the test should be completely listed.

Author Response

Reviewer 3

Review Comments:

Thanks for your appreciation. We have reviewed the manuscript point by point according to the reviewer.

1、  The abstract does not state what value the study of the unique properties of Liquor-based chutneys provides, only the background, methodology, content, and results of the study, which should be supplemented with its research implications.

Thank you very much. The abstract was rewritten and included the main results of this research.

2、  The background information on the hot sauce is adequate and raises the status of issues such as the lack of novel, non-volatile characteristics of hot sauce products, but does not clearly answer this and should be supplemented with targeted opinions.

Peppers come from Magic Plant Farms.

https://magicplantfarms.com

Tequila comes from Republic National Distributors. It’s 80 proof, and we use it right out of 1.75 L bottles.

All other liquors come from UltraPure.

https://bulkalcohol.com

They ship to be us at high proof to save on the weight of shipping. We dilute everything to 80 proof before mixing. We use Baton Rouge City water. We maintain the city’s Quality Report in our records.

Our bourbon comes to us at 117 proof. Our rum comes at 150 proof, and our vodka comes as 190 neutral grain alcohol.

Our xanthan gum is supplied by CP Kelko.

https://www.cpkelco.com/products/xanthan-gum/

We use the Keltrol type of gum.

3、  The paper characterizes vinegar-based hot sauces and Liquor-based hot sauces by determining the number of indicators such as physicochemical properties, volatile compounds, microbiological quality, hedonic scores, emotions, and purchase intent to illustrate the unique properties of wine-based hot sauces. The research methodology is presented and the article is well laid out and structured to illustrate the unique properties of wine-based hot sauces. However, the results of the paper are imperfect, and what advantages its unique attributes have, for example, how volatiles affect key product attributes are not specifically written, and it is recommended that the flavor characteristics of volatiles be added.

Dear reviewer, thank you very much for your appreciation. The aromatic attributes provided by the different volatile compounds have been included in the manuscript. The sensory attributes provided by the different volatile compounds identified were: dimethyl ether, fait ethereol odor; ethanol have alcohol odor; butanoic acid 3-methyl-hexyl-ester present odor of wine and ethyl-9-decenoate, 4-methylpentyl-3-methylbutanoate, and hexyl n-valerate preset green and fruity odor.

4、  The chart is clear and reliable, but the literature citation is old, less in the past 5 years, and the sample reagents, instruments, and equipment used in the test should be completely listed.

Thank you very much. In the methodology, the brand and model equipment used, as well as the reagents used in each case, are specified. The references are old, as they are standardized analysis methods and are still in use.

Round 2

Reviewer 1 Report

After taking a look on the revised text I still stand by my opinion that I made after the first review of the manuscript – the manuscript should be rejected. The impression is that the authors did not adequately respond to the comments I made on the first version. If you take a look at my comments, my opinion was that the manuscript did not deserve even a major revision and suggested it should be rewritten and submitted as a new one elsewhere. The text changes that the authors made (at least in the text up to the Results and Discussion chapter) are minor without addressing the main objections (e.g. they just skipped the part within the ‘2.6 Sensory Evaluation’ related to the type of scale used for evaluating emotional effects – it cannot be hedonic scale!!!! and this is not a minor mistake to measure a variable with the scale tailored for measurements in completely different dimension - these results are not valid at all). So, for sure I will not waist the time again explaining on two pages why I think that the manuscript is not scientifically sound enough. My opinion is that the manuscript should be rejected.

Author Response

Reviewer 1
Comments to Author

The manuscript represents a study of potential using of different types of spirits (tequila, rum, vodka, bourbon) instead of traditional base-ingredients in the production of hot pepper sauces, with an intention of bringing some links between certain objective quality parameters and some factors related to consumers (including their acceptance, emotional responses and purchase intent).

Thanks for your appreciation. We have reviewed the manuscript point by point according to the reviewer.

Review Comments:

  1. Introduction:

The main topic of the manuscript is the use of certain spirits in the production of hot  sauces, but there is no word about that within the Introduction part. Even if this was some kind of pioneer work (which I doubt), something should be written about possibilities and justification of using spirits instead of traditional base-ingredients.

Thank you very much for your appreciation. It is a pioneering and innovative work where traditional ingredients are replaced by spirits

  1. Materials and Methods:

Subchapters 2.1 (Experimental Design) and 2.2 (Accelerated shelf-life study):

The experimental design is completely unclear to the reader. It is not even clear (by reading M&M) how many experimental samples were used in the study.

Thank you very much for your support. A flow diagram has been made in which the experimental design is explained. The abbreviation M&M was withdrawn for being confusing.

One can conclude that the shelflife study was performed in accelerated conditions, but taking the way this was written in the manuscript it is not clear what conditions were exactly applied (real duration of storage at what temperature) and what periods within the storage were taken for sampling. For example, in L92-93 it says that “each day of storage at 40°C was considered as, at least, 4 days at ambient temperature”, and taking into account that “volatile compounds and sensory analysis were done after day 1 of storage” (L79-80), it is not clear whether this one day was a real storage day ataccelerated conditions, or a day at ambient conditions.

Thank you very much. Under accelerated conditions, the physicochemical properties were measured. Volatiles were not measured under accelerated conditions, they were measured at storage temperature which was 40°C.

The “rule of ten” should be referenced, and better explained in the manuscript. The sentence in L90-93 is also unclear (please read it by yourself). “Thus, considering the ambient temperature around 20°C represents Q10 of 2 and thatin this study the storage temperature of the hot sauce was 40°C, then there are two cycles, 22, and therefore each day of storage at 40°C was considered as, at least, 4 days at ambient temperature”. What two cycles? What does the number 22 mean? So, the only conclusion the reader can draw is that the experimental design is not explained at all.

Dear editor, thank you very much. The 22 is a formatting error. These are 2 cycles that are 4 days long. The reference used in this methodology is number 6) Rodiles-López, J.O.; Garcia-Rodriguez, D.A.; Gomez-Orozco, S.Y.; Tiwari, D.K.; Coria-Téllez, A. V. Food quality evaluation of accelerated shelf life of chili sauce using Fourier transform infrared spectroscopy and chemometrics. J. Food Process. Preserv. 2020, 44, e14350.

Subchapter 2.5 (Rheological measurements):

Within rheological measurements it says that “three types of experiments were performed” 2 (L126-127), and also that “all experiments were conducted at 25 °C” (L128). So, should it be “measurements” instead of “experiments”? And if not, what kind of experiments were all about, and how does the temperature of 25 °C fit into the experimental design (not)explained in chapter 2.1?

Thank you very much. It has been corrected in the manuscript. 3 types of analysis have been carried out, instead of 3 experiments and all of them at a temperature of 25 ºC

Subchapter 2.6 (Sensory evaluation):

L143-144: “The samples were analyzed using a three-digit code after week 1 of storage.” (???). How could the samples be analyzed by using a three-digit code, at all? OK, I assume that this was some kind of language error. But again, here it says “after week 1 of storage”, and in Ch. 2.1 “after day 1 of storage”. So, when exactly, day 1 or week 1? This brings us back to the comments I posed in previous text regarding the Ch. 2.1 and 2.2.

Each sample has 3 digits which is a standard method to perform sensory analysis. Sensory analyzes were performed one day after

L152-153: “The sensory study plan was executed using a Counterbalance Design (t = 5, b = 2, k= 40) and analyzed f day after the production of samples”. If this was an attempt to explain the sensory study plan, it is not clear to the reader again. I assume that this was some kind of balanced incomplete block design, but hey, these indicator marks, namely t, b, k, f, are not universal, one can use any letter from alphabet, so an explanation is missing what exactly do thet = 5, b = 2, k = 40 and f mean.

Thank you very much for the appreciation. The nomenclature used was: b=2 means that a panelist only tested two samples, since they are hot sauces and cannot bear to test more than two samples. t=5 relates to the number of treatments including the control and k=40 means that each block was tasted by 40 people.

Subchapter 2.8 (Statistical analysis):

L165-167: “Bonferroni (Dunn) was used to determine significant differences for main effects (liquor type and storage time), interaction effect (liquor type × storage time), and hedonic responses”. Dunn’s test is a post hoc multiple comparison test which can be done after e.g. ANOVA, or in adjusted form after ANCOVA, but this sentence again makes me confused. I will repeat: .... significant differences for (1) main effects, (2) interaction effect, and (3) hedonic responses. (???) This doesn’t make a sense.

Thank you very much for your appreciation. It HAS been rewritten in the manuscript. To see differences in the regressions, the anvo proamixed was used. The hedonic scale is analyzed with an ANOVA with generalized linear model.

Reviewer 3 Report

Current manuscript can be accepted

Author Response

Dear Reviewer:

Thank you very much for reviewing the manuscript and for your contributions.

Sincerely,

PhD Ismael Montero